**www.cambridge.org/qrd**

# Unzipping of knotted DNA via nanopore translocation

Antonio Suma[1,2] and Cristian Micheletti[3]

[1]Dipartimento di Fisica, Università di Bari and INFN, Sezione di Bari, Bari, Italy; [2]Institute for Computational Molecular Science, Temple University, Philadelphia, PA, USA and [3]Physics Area, Scuola Internazionale Superiore di Studi Avanzati (SISSA), Trieste, Italy

DNA; knots; nanopore translocation; topological friction; unzipping

**Corresponding author:**
Cristian Micheletti;
Email: michelet@sissa.it

## Abstract

DNA unzipping by nanopore translocation has implications in diverse contexts, from polymer physics to single-molecule manipulation to DNA–enzyme interactions in biological systems. Here we use molecular dynamics simulations and a coarse-grained model of DNA to address the nanopore unzipping of DNA filaments that are knotted. This previously unaddressed problem is motivated by the fact that DNA knots inevitably occur in isolated equilibrated filaments and *in vivo*. We study how different types of tight knots in the DNA segment just outside the pore impact unzipping at different driving forces. We establish three main results. First, knots do not significantly affect the unzipping process at low forces. However, knotted DNAs unzip more slowly and heterogeneously than unknotted ones at high forces. Finally, we observe that the microscopic origin of the hindrance typically involves two concurrent causes: the topological friction of the DNA chain sliding along its knotted contour and the additional friction originating from the entanglement with the newly unzipped DNA. The results reveal a previously unsuspected complexity of the interplay of DNA topology and unzipping, which should be relevant for interpreting nanopore-based single-molecule unzipping experiments and improving the modeling of DNA transactions *in vivo*.

## Introduction

A series of advancements in pore translocation setups have brought this single-molecule technique to the forefront of numerous applications, far exceeding the originally envisioned purpose of sequencing nucleic acids (Kasianowicz *et al.*, 1996; Palyulin *et al.*, 2014; Deamer *et al.*, 2016). Recent applications include advanced molecular sensing (Rahman *et al.*, 2019; Wang *et al.*, 2021; Leitao *et al.*, 2023), out-of-equilibrium stochastic processes (Kantor and Kardar, 2004; Grosberg *et al.*, 2006; Sarabadani and Ala-Nissila, 2018; Suma *et al.*, 2023), RNA unfolding (Bandarkar *et al.*, 2020; Suma *et al.*, 2020), protein sequencing (Asandei *et al.*, 2020; Yu *et al.*, 2023), and probing of intra- and inter-molecular entanglement (Huang and Makarov, 2008; Rosa *et al.*, 2012; Suma *et al.*, 2015; Narsimhan *et al.*, 2016; Plesa *et al.*, 2016; Suma and Micheletti, 2017; Marenda *et al.*, 2017; Caraglio *et al.*, 2017; Weiss *et al.*, 2019; Caraglio *et al.*, 2020; Rheaume and Klotz, 2023).

One of the most exciting avenues for nanopore translocation is probing the structure and function of biological polymers. A notable example is offered by exonuclease-resistant RNAs (xrRNAs) (Pijlman *et al.*, 2008; Chapman *et al.*, 2014; Akiyama *et al.*, 2016; MacFadden *et al.*, 2018; Slonchak *et al.*, 2020; Vicens and Kieft, 2021). These modular elements, consisting of only a few dozen nucleotides, are located at the 5′ end of the RNA genome of flaviviruses and are responsible for infections such as Zika, dengue, and yellow fever (Slonchak *et al.*, 2018). xrRNAs are distinguished by their unique and diverse functional responses when pulled through the lumen of enzymes that process nucleic acids. Specifically, xrRNAs resist degradation by exonucleases that translocate nucleic acids from the 5′ end. However, they can be processed by replicases and reverse transcriptases, which translocate RNAs from the 3′ ends.

A mechanistic explanation for this behavior was provided by the theoretical and computational study of Suma *et al.* (2020), where a pore translocation setup, mimicking the action of processive enzymes, was used to unzip xrRNAs from both ends. The study, further supported by later work (Becchi *et al.*, 2021; Niu *et al.*, 2020), reported that the short and yet heavily entangled structure of xrRNAs, which includes several pseudoknots (Akiyama *et al.*, 2016), contributes to a strongly directional translocation response. Pulling xrRNAs from the 5′ end causes the molecule to close in on itself and resist further unzipping, explaining its resistance to exonucleases; conversely, when translocated from the 3′ end, the molecule unravels progressively, explaining its processability by replicases and helicases/reverse transcriptase (Suma *et al.*, 2020).

Differently from RNAs, double-stranded DNA (dsDNA) filaments are usually well described by general polymer models with torsional and bending rigidity (Chirico and Langowski, 1994; Klenin *et al.*, 1998; Vologodskii and Cozzarelli, 1994). Although dsDNA does not form the complex architectures typical of RNAs, it can become knotted due to its spontaneous dynamics,

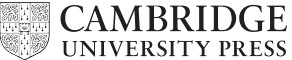

both in bulk and under confinement (Rybenkov *et al.*, 1993; Arsuaga *et al.*, 2002; Marenduzzo *et al.*, 2009). Additionally, dsDNA filaments can become knotted through the actions of type II topoisomerases, which perform strand crossings that can potentially alter the topological state of DNA, establishing a homeostatic level knotting that needs to be tightly regulated to avoid detrimental consequences for living cells (Portugal and Rodríguez-Campos, 1996; Rybenkov *et al.*, 1997; Olavarrieta *et al.*, 2002; Deibler *et al.*, 2007, p. 1; Valdés *et al.*, 2018; Valdés *et al.*, 2019).

The emergence of DNA knots, be they formed spontaneously or introduced by topoisomerases, has been traditionally based on gel electrophoresis (Dröge and Cozzarelli, 1992; Trigueros *et al.*, 2001; Valdés *et al.*, 2019). Such setups harness the different hindrances experienced by molecules with different knot types when moving through the gel mesh. Its main limitation regards the maximum length to which it can be practically applied, which is of the order of 10 kb.

Recent breakthroughs have opened the possibility of overcoming this practical limit by resorting to pore translocation setups (Plesa *et al.*, 2016; Suma and Micheletti, 2017; Sharma *et al.*, 2019). Suitable choices of the pore diameter allow for translocating the DNA knots and reveal their passage from the drop of the ionic current, which depends on the obstruction of the pore caused by the passing knotted region and involves at least three dsDNA strands. While the technique may not be sensitive to the knot type and knot size (Suma and Micheletti, 2017), it allows for probing the so-called topological friction (Rosa *et al.*, 2012; Suma *et al.*, 2015). The latter can be revealed by using pores sufficiently narrow that only one dsDNA filament can pass through, causing the knot to remain localized at the pore entrance, hindering the translocation of the remainder of the filament that has to slide along the contour of the knotted region to pass through. In such a setup, the hindrance to translocation can depend on the knot type and the driving force (Rosa *et al.*, 2012; Suma *et al.*, 2015; Narsimhan *et al.*, 2016). Increasing the driving force makes the knots tighter, enhancing the friction to the point that the translocation process can even be stalled indefinitely, as illustrated in Figure 1, which presents results from simulations specifically carried out for this study.

At the same time, dsDNA typically undergoes another type of *in vivo* transaction operated by, for example, helicases, namely unzipping. In the pore translocation setup, this effect can be mimicked by reducing the pore diameter so that only one strand of the

DNA duplex can pass and is harnessed for fast and reliable genome sequencing (Manrao *et al.*, 2012; Jain *et al.*, 2018). This interesting out-of-equilibrium setup has been used before to explore fundamental aspects of the equilibrium thermodynamics (Dudko *et al.*, 2008), from the sequence-dependent free energy profile (Huguet *et al.*, 2010) of unzipping to base pairing (Suma *et al.*, 2023) to the dynamical regimes appearing at different forces (Suma *et al.*, 2023), which differ considerably from those occurring without unzipping both in terms of typical translocation times and scaling behavior (Palyulin *et al.*, 2014; Chen *et al.*, 2021; Suma *et al.*, 2023).

The examples above underscore three key points. First, the structural features of nucleic acids include physical entanglements, which can have complex and significant functional reverberations *in vivo.* Second, pore translocation setups are indispensable tools for mimicking the action of enzymes and probing the structural response of nucleic acid tangles at the single-molecule level. Third, the external control afforded by translocation setups, such as varying pore size and force application protocols (constant, time-ramped, oscillating), provides an ideal context for understanding the microscopic basis of the observed unzipping responses. This understanding offers crucial clues for decoding how nucleic acid architecture informs translocation.

One open problem that intersects all three aspects above is understanding how the statistically inevitable presence of knots can interfere with DNA unzipping by translocation. Studies have yet to be conducted on this process, which is qualitatively different from translocating knotted DNA without unzipping. For this reason, the insights gleaned from the pore translocation of knotted DNA cannot be directly applied to the unzipping scenario. This leaves fundamental questions about the unzipping of knotted dsDNA unanswered, such as: (i) how large must the driving force be to keep the knot tight at the pore entrance and prevent it from diffusing along the chain, (ii) what is the force-dependent topological friction, and (iii) how does this friction depend on the type of knot? These questions have implications also for *in vivo* DNA processing by enzymes, given that DNA knots not removed by defective topoisomerases can stall such processes, with negative consequences for the cell (Shishido *et al.*, 1987; Postow *et al.*, 2001; Olavarrieta *et al.*, 2002; Deibler *et al.*, 2007, p. 44; Valdés *et al.*, 2018; Valdés *et al.*, 2019). Although the interplay of DNA topology and unzipping is recognized as a key element of *in vivo* DNA transactions, the detailed characterization of the process has so far

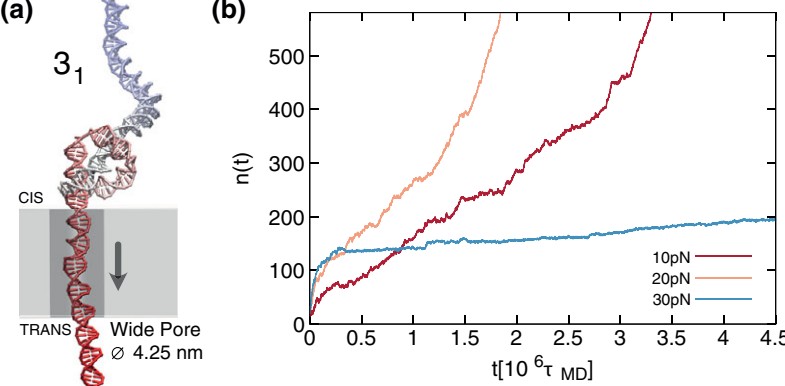

**Figure 1.** (a) Snapshot of a trefoil ($3_1$) knotted dsDNA translocating through a wide pore with a 4.25 nm diameter, allowing for the passage of a single double strand, thus blocking the knot. The total applied translocating force is 30 pN, sufficient to maintain the knot in a tight state near the pore entrance. (b) Time evolution of the number of base pairs, *n*, which have translocated from the *cis* to the *trans* side of the slab where the nanopore is embedded. The trajectories are for a $3_1$-knotted dsDNA chain at three different driving forces. The translocation process speeds up when *f* is increased from 10 to 20 pN and then slows down, and even stalls, at higher forces due to the topological friction in the tightened knotted region.

remained beyond the scope of single-molecule manipulation experiments.

Here, we address these questions with molecular dynamics simulations of a coarse-grained DNA model, oxDNA2 (Ouldridge *et al.*, 2011; Snodin *et al.*, 2015). We first consider the reference case of the nanopore unzipping of unknotted DNAs and study their translocation compliance at different forces. Next, we turn to knotted DNAs and discuss how the unzipping speed varies with knot type and applied force. Finally, we address the complementary aspect, namely how unzipping by translocation affects the knotted region, particularly its length and contour dynamics.

Notably, we do not observe significant effects related to knots at pulling forces of 50 pN, which is of the same order as the forces that can be generated by molecular motors (Smith *et al.*, 2001). The results are suggestive that topological entanglement may not significantly interfere with *in vivo* DNA unzipping operated by enzymes. However, the interplay of topology and unzipping is significantly different at 100 pN and larger forces, with major effects on the translocation process and knot sliding dynamics.

## Results

To study the nanopore unzipping of knotted DNA filaments, we applied Langevin molecular dynamics simulations to 500-bp long DNA filaments described with the oxDNA2 model (Ouldridge *et al.*, 2010, 2011; Snodin *et al.*, 2015), a coarse-grained DNA representation with interactions parameters tuned to reproduce phenomenological data for DNA properties and interactions, including base pairing, stacking, and twist-bend couplings. The model's predictive capabilities were validated in a variety of contexts, including the application of external mechanical forces (Romano *et al.*, 2013; Matek *et al.*, 2015; Mosayebi *et al.*, 2015; Engel *et al.*, 2018).

The initial states were prepared from five different equilibrated (Monte Carlo generated) conformations of the 500 bp filaments. The five conformations were all unknotted because the 500 bp contour length, corresponding to about 10 DNA persistence lengths, is too short for significant spontaneous knotting in equilibrium (Rybenkov *et al.*, 1993; Tubiana *et al.*, 2013; Uehara *et al.*, 2019). The 500-bp long filaments were next attached to leads that consisted of a double-stranded knotted region with $3_1$, $4_1$, and $3_1\#3_1$ topology – the knotted region was omitted for unknotted ($0_1$) case – plus a 40-base long single-stranded stretch, pre-inserted into a pore (see Figure 2). The translocation process was driven by pulling the nucleotides inside the pore with a total longitudinal force, $f$ of 50, 100, and 150 pN. The pore diameter, 1.87 nm, was

chosen small enough that only a single DNA strand can pass through it, causing translocating DNAs to unzip.

### Nanopore unzipping of unknotted DNA

Figure 3a illustrates, for reference, the translocation response of unknotted DNA filaments. The traces show the number of translocated nucleotides as a function of time, $n(t)$, for five independent trajectories at each indicated force. Note that traces start at about 40, corresponding to the length of the single-stranded DNA (ssDNA) segment of the lead that is already threaded inside the pore at $t = 0$.

The traces at $f = 50$ pN have an overall linear appearance, indicative of an approximately constant unzipping velocity. However, the traces at the two largest forces, 100 and 150 pN, deviate noticeably from linearity. The convexity, or upward curvature of the late part of traces ($n(t) > 300$), indicates that the average translocation speed increases in the second half of the translocation.

The translocation/unzipping speeds vary significantly across the forces. For comparison, average translocation times were computed at the 400 translocated bases mark, a convenient reference given the graphs' range in Figure 3. The average times are equal to $3.0 \cdot 10^6, 6.9 \cdot 10^5$ and $3.2 \cdot 10^5 \tau_{MD}$ for $f = 50$, 100, and 150 pN, respectively. In particular, we note that the above translocation/unzipping times do not follow the inverse force relationship expected for simple dissipative processes. Specifically, a twofold force variation from 50 to 100 pN produces an order-of-magnitude change in unzipping time.

The results parallel and expand those reported in Suma *et al.* (2023), where data for the out-of-equilibrium unzipping process of dsDNA were used within a theoretical framework that enabled reconstructing the free-energy profile of single base-pair formation. In that context, it was found that the unzipping process proceeded at relatively constant velocity for forces below $\sim 60$ pN and could be modeled as a drift-diffusive process. At the same time, progressive speed-ups during translocation were observed at larger forces associated with an anomalous dynamics regime. By modeling the unzipping as a stochastic process in a one-dimensional tilted washboard (periodic) potential, it was shown that 60 pN force corresponded to lowering the barrier to unzip a base-pair to a value where advective transport becomes relevant over diffusion (Suma *et al.*, 2023). Additionally, we recall that DNA undergoes significant structural deformations, that is, overstretching, at about this same force when mechanically stretched (Smith *et al.*, 1996), and that the oxDNA2 model inherently accounts for these effects (Romano *et al.*, 2013). Thus, the crossover from linear to non-linear

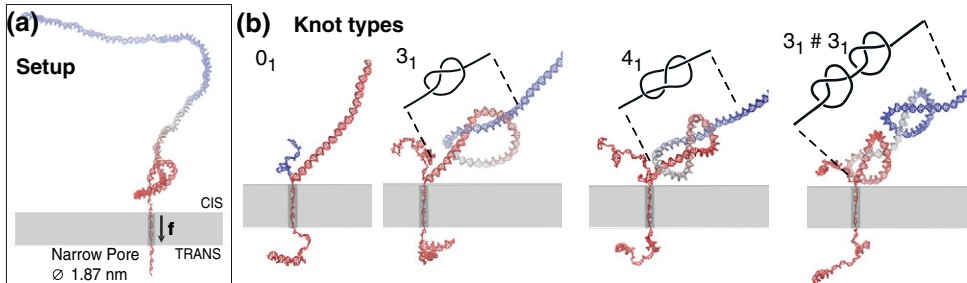

**Figure 2.** (a) Schematic illustration of the initial setup: an unknotted, equilibrated filament is attached to a lead consisting of a tightly-knotted double-stranded segment plus a single-stranded one pre-inserted into a cylindrical pore embedded in a slab. The narrow pore has a diameter of 1.87 nm, allowing only a single DNA strand to pass at a time. (b) Configurations of 500 bp-long DNA filaments during the simulated translocation-driven unzipping. The four snapshots are close-ups of the system near the pore and illustrate the different considered topologies: unknot ($0_1$), trefoil ($3_1$), figure-of-eight ($4_1$), and the composite granny knot ($3_1\#3_1$).

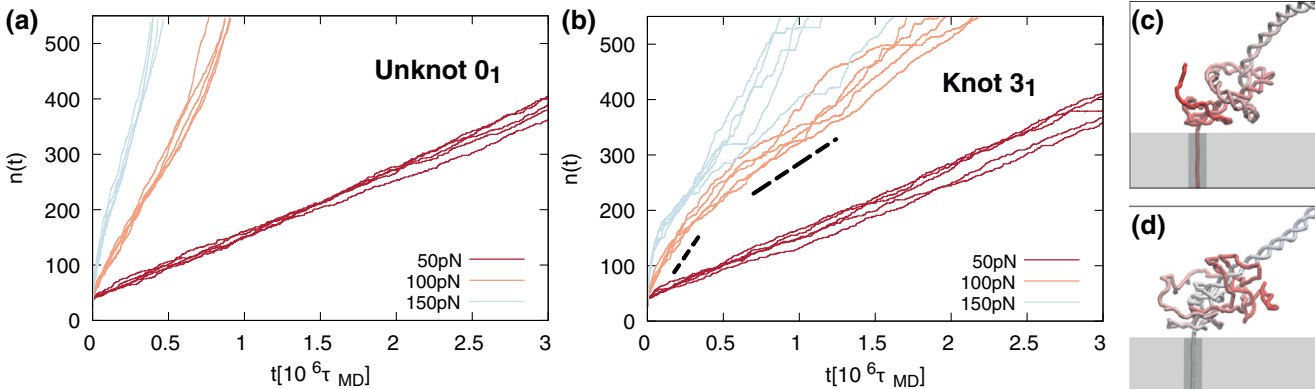

**Figure 3.** Number of translocated nucleotides, $n$, as a function of time, $t$ for dsDNA filaments that (a) are unknotted and (b) have a $3_1$ knot; see Figure 2 and methods. The traces are for pulling forces of 50, 100, and 150 pN, with five independent trajectories for each case. The dashed lines highlight two distinct velocity regimes in the 100 pN trajectories, a feature also present in some of the 150 pN traces. Configurations in panels (c) and (d) are snapshots at 100 pN for the $3_1$ knot taken before and after the change in regime.

translocation/unzipping observed upon increasing $f$ from 50 to 100 pN is consistent with other qualitative changes of DNA properties in the same force range.

### Nanopore unzipping of $3_1$-knotted DNA

The force-dependent translocation response is dramatically changed when the unknotted lead is replaced by a knotted one, even when the topology is the simplest non-trivial one. This emerges by inspecting Figure 3b, which shows the unzipping traces for DNA strands starting with a moderately tight trefoil-knotted ($3_1$) lead.

The comparison of the two panels in Figure 3 clarifies that at $f = 50$ pN, the unzipping of knotted and unknotted chains proceed almost undistinguishably. The average unzipping velocities of the two sets of traces, measured as nucleotides translocating per unit time, are compatible with statistical uncertainty, $1.309 \pm 0.028 \cdot 10^{-4}\tau_{MD}^{-1}$ for the $0_1$ topology and $1.288 \pm 0.036 \cdot 10^{-4}\tau_{MD}^{-1}$ for the $3_1$ case. The main perceived difference is the spread of the five traces, which is larger for the knotted cases.

However, increasing the force to 100 pN or more causes the unzipping of knotted chains to proceed more slowly and heterogeneously than unknotted DNAs. For $f = 100$ pN, the relative slowing down of the average velocity is approximately twofold, and the same holds for the largest considered force, $f = 150$ pN.

In addition, two different regimes are discernible, highlighted by the dashed lines for the $f = 100$ case, with snapshots before and after the change in regime presented in Figure 3c and d. Initially, the trefoil-knotted filament unzips at the same rate as the unknotted ones. Beyond this regime, which applies to the first 200 bp, the process slows down noticeably while also becoming more heterogeneous. An analogous effect is found for the $f = 150$ pN case, but with the important difference that the transient where the velocity is the same as in the unknotted case has a shorter duration and covers fewer base pairs (150). As we discuss later, the change in velocity is a consequence of the force-induced tightening of the knot near the pore entrance, which adds a significant hindrance – also termed topological friction – to the translocation process.

### Effect of knot topology on DNA unzipping

We additionally considered leads with figure-of-eight ($4_1$) and granny ($3_1\#3_1$) knots to extend the range of topological complexity beyond the trivial ($0_1$) and trefoil ($3_1$) knot types. As a conventional measure of knot complexity, we consider the crossing number, corresponding to the minimum number of crossings in the simplest possible non-degenerate projection. This complexity measure equals 0, 3, 4, and 6 for the $0_1$, $3_1$, $4_1$, and $3_1\#3_1$ knots, respectively.

The unzipping traces for all topologies are shown in Figure 4. We stress that we purposely attached the same set of equilibrated 500-bp long dsDNA conformations to the battery of differently knotted leads. With this choice, emerging systematic differences across the different topologies can be directly ascribed to the different knotted states of the lead and not to other effects, such as the initial DNA conformation on the *cis* side.

The data in panel (a) show that all traces are well-superposed and consistent with an approximate linear (constant velocity) behavior at the lowest considered force, $f = 50$ pN. This result confirms the earlier observation that the unzipping response is mainly independent of the knotted state at sufficiently small $f$ (Figure 3).

The data in panels (b) and (c), which refer to $f = 100$ and 150 pN, respectively, are consistent with those of the trefoil knot case (Figure 3), too, in that the unzipping proceeds practically identically for all topologies of an initial tract, which spans 200 bp at $f = 100$ pN and 100 bp at $f = 150$ pN. Beyond this point, the unzipping slows down for all non-trivial knot types. At $f = 100$ pN, we observe that the highest unzipping hindrance is offered by the $4_1$ knot, followed by the composite $3_1\#3_1$ knot, and the $3_1$ and $0_1$ topologies. We recall that $3_1\#3_1$ knot has the highest nominal complexity in the considered set, and yet it is not associated with the slowest unzipping at $f = 100$ pN, which is noteworthy. However, at 150 pN, the $3_1\#3_1$ and $4_1$ knots offer comparable hindrance, while the unzipping of the $3_1$ case is faster and that of the unknot $0_1$ remains the fastest.

The findings can be interpreted in terms of previously published results on the translocation – without unzipping – of knotted chains of beads (Suma *et al.*, 2015). For such a system, it was shown that each prime knotted component behaves as a dissipative structural element that interferes with the mechanical tension propagating to the chain remainder by significantly reducing it. Without unzipping, the translocation velocity for the case of concatenated trefoil knots ($3_1\#3_1$) was mainly defined by the force dissipation within the first $3_1$-knotted component, which is less complex than the $4_1$ knot. This observation helps rationalize that in specific force regimes, the hindrance of the $3_1\#3_1$ case can be intermediate to the $3_1$ and $4_1$ ones.

The results of Figures 3 and 4 establish two points. First, the effects of DNA knots on the unzipping process are negligible, up to forces of at least 50 pN. This is a relatively large force for practical

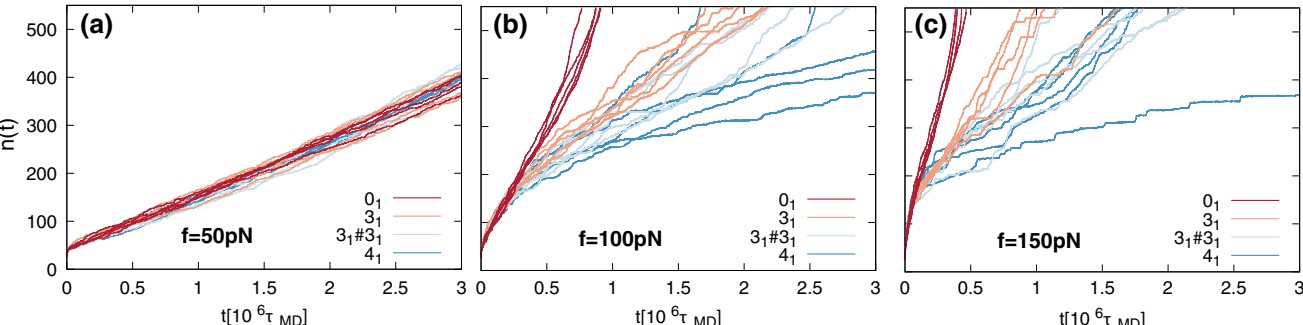

**Figure 4.** Number of translocated nucleotides, $n$, as a function of time, $t$, for DNA filaments with different knot types and at different driving forces, as indicated. The traces of five independent trajectories are shown for each case.

and biological purposes in that it is comparable to the force generated by the most powerful molecular motors (Smith *et al.*, 2001), and corresponds to the onset of the DNA overstretching transition observed in force spectroscopy (Smith *et al.*, 1996). Second, at forces of 100 pN and beyond, the presence of knots is associated with significant slowing downs of the unzipping process depending on the interplay of knot topology and driving force.

### Effect of the unzipped strand interfering with the knot

A noteworthy aspect of Figure 4 is the noticeable heterogeneity of the unzipping traces at $f = 100$ and 150 pN. For instance, over the five $4_1$ traces collected at $f = 100$ pN, the time required to reach the $n(t) = 400$ mark can range from $1.2 \cdot 10^6 \tau_{MD}$ to $3.4 \cdot 10^6 \tau_{MD}$, a threefold ratio. For comparison, at $f = 50$ pN, the same ratio is only 1.02.

Visual inspection of the unzipping trajectories revealed that the heterogeneity is not only due to the presence of the knot but also to the hindrance arising from the unzipped ssDNA strand on the *cis* side becoming entangled with the knotted region. The effect is illustrated in Figure 5, which presents typical DNA conformations on the *cis* side of the pore.

As illustrated, the knotted region typically leans against the pore entrance at the smallest considered force, $f = 50$ pN. However, at $f = 100$ pN and 150 pN, the knot is often not in direct contact with the pore but is kept at a finite distance from it by the *cis* unknotted strand that wraps around the dsDNA stem immediately below the knot. These wrappings arise from the torsional

stress generated by the unzipping of double-helical DNA (Fosado *et al.*, 2021). When the stress is generated faster than it can be dissipated (Zheng *et al.*, 2024), it can cause the relative rotation of the newly-unzipped and yet-to-unzip DNA strand, and hence their wrapping.

Like those of Figure 5, the wrapped conformations inevitably offer a multi-tier hindrance to nanopore unzipping. The translocating dsDNA experiences the combined friction from the knot and the wrapped unzipped filament to a degree that depends on the tightness and number of turns of the latter, thus increasing the heterogeneity of the unzipping process.

### Knot dynamics

We next considered the sliding dynamics of the knots along the *cis* portion of the DNA chain, which we addressed by tracking in time the nucleotide indices corresponding to the two ends of each knot. We employed the method of Tubiana *et al.* (2011), which uses a bottom-up search scheme to identify the shortest segment of a chain that, once closed with a suitable arc, yields a ring with the sought knot topology (Tubiana *et al.*, 2018).

Figure 6a illustrates the typical evolution of the contour positions of $3_1$, $4_1$ $3_1\#3_1$ knots for different forces. As indicated in the accompanying sketches, the $n_1$ and $n_2$ traces indicate the nucleotide indices of two ends of $3_1$ and $4_1$ knots and of the first (pore proximal) component of the $3_1\#3_1$ composite knot. The indices for the second component of the composite knot are instead indicated as $n_3$ and $n_4$. Additionally, the plots in

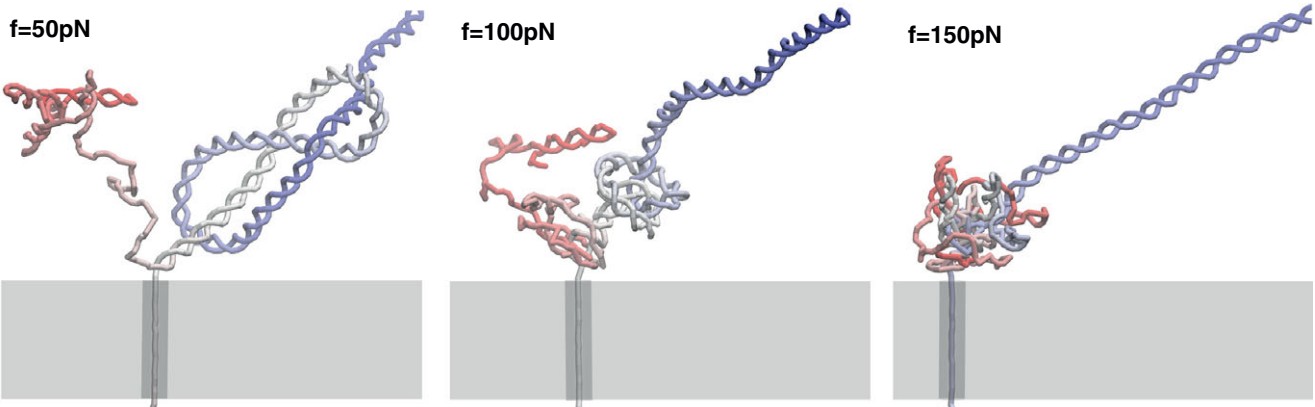

**Figure 5.** Typical conformations of a $4_1$-knotted dsDNA filament at intermediate stages of translocation and increasing driving force, 50, 100, and 150 pN. At the two largest forces, one observes knot tightening and the wrapping of the *cis* unzipped strand around the dsDNA region proximal to the pore.

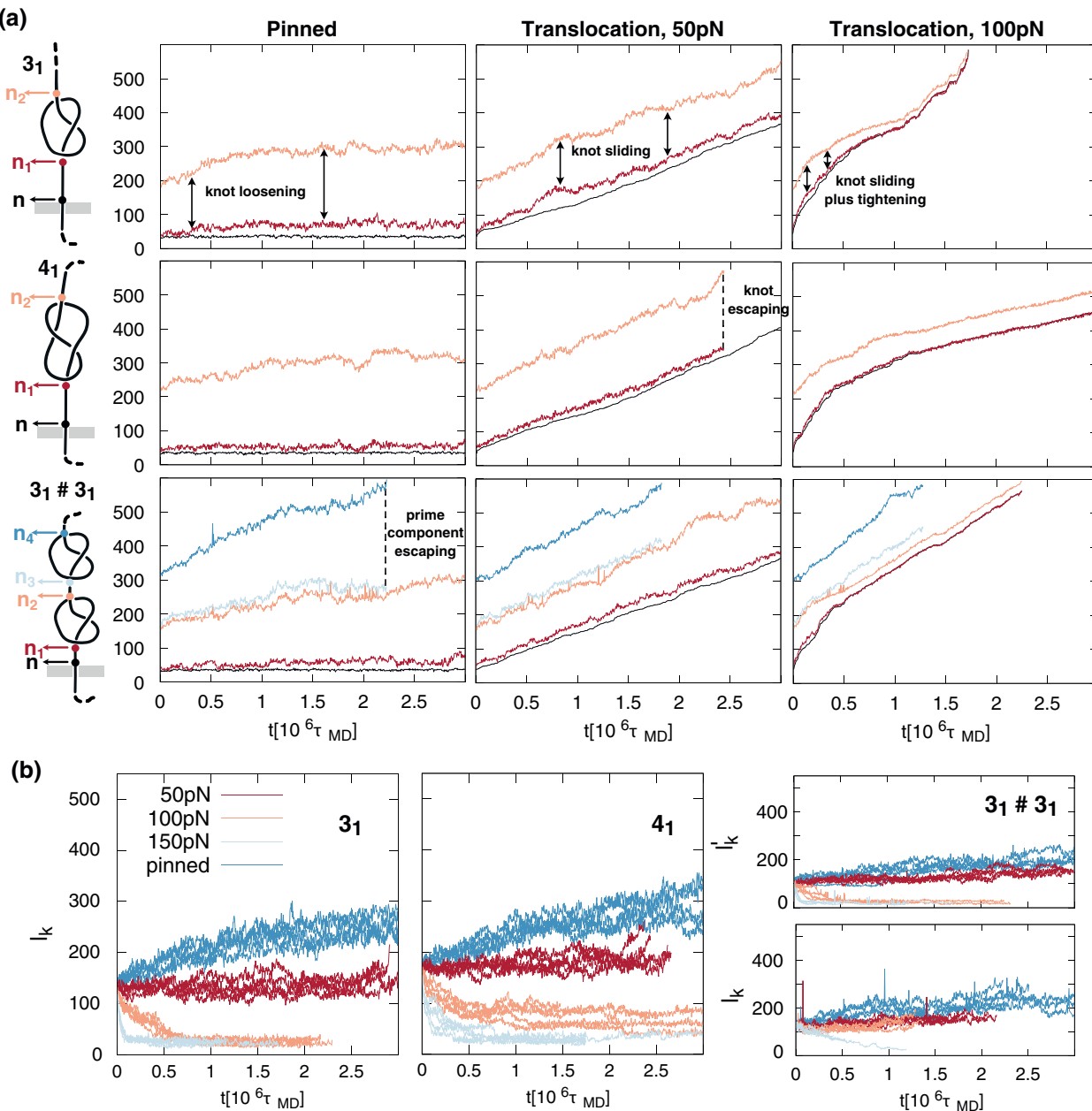

**Figure 6.** (a) From top to bottom, three rows show the typical evolution of the contour positions of $3_1$, $4_1$, and $3_1\#3_1$ knots in different setups. Sketches on the left provide the legend for the plotted nucleotide indices corresponding to the knot ends, $n_1, n_2$ for $3_1$ and $4_1$ knots, and $n_1, n_2, n_3, n_4$ for the $3_1\#3_1$ knot. The $n(t)$ trace marks the index of the nucleotide at the pore entrance (or, equivalently, the number of translocated nucleotides, as in previous figures). The first column is for a setup where a base inside the pore is kept pinned. The second and third columns represent translocation cases at 50 and 100 pN, respectively. The traces in panel (b) illustrate the time evolution of the knot length, $l_k = n_2 - n_1$, for $3_1$, $4_1$ topologies, and for each of the two prime components for the $3_1\#3_1$ topology, $l_k = n_2 - n_1$ and $l'_k = n_4 - n_3$. The knot ends for prime and composite knots were detected using the software KymoKnot (Tubiana *et al.*, 2018, see Methods). Each plot shows the pinned case, as well as 50, 100, and 150 pN pulling forces. The traces of five independent trajectories are shown for each case.

Figure 6a show the traces of the index of the nucleotide at the pore entrance, $n$.

The data in Figure 6a allows for tracking various quantities of interest as a function of time, $t$. For instance, $n(t)$ is directly informative of the progress of the translocation/unzipping process. In contrast, the contour distance $n_1(t) - n(t)$ conveys how much the knotted region stays close to the pore during unzipping. In addition, the contour lengths of the prime knotted components are given by $l_k = n_2(t) - n_1(t)$ and $l'_k = n_4(t) - n_3(t)$ and are shown in Figure 6b for the five independent trajectories of the considered cases.

## Knot evolution in pinned DNA chains

The first column in Figure 6a is for the case where the ssDNA end inserted in the pore is not subject to a translocating force but is held in place by pinning a nucleotide inside the pore. The evolution of the pinned knotted configurations covers a time span of $3 \cdot 10^6 \tau_{MD}$, comparable to the typical duration of unzipping processes at 100 pN. This case serves as a term of reference. Specifically, it establishes how the knotted DNA region evolves from its initial moderately tight state in the presence of the pore and slab but without any interference from a concurrent translocation/unzipping process and without mechanical tension propagating from the

pore. The traces of the pinned case show a systematic increase in knot lengths across all three considered topologies; see also Figure 6b for $l_k$ and $l'_k$. The progressive loosening of knots reduces the system's bending energy compared with the initial state, where knotted components are moderately tight ($\sim$150 bp) and significant curvature is thus packed into relatively short dsDNA stretches. The expansion of the knot is visibly asymmetric at the two ends because the knot cannot penetrate inside the pore and can only expand on the *cis* side.

The evolution of the $3_1\#3_1$ case in the first column of Figure 6a is particularly interesting. The expansion is slowest for the first component (the one proximal to the pore), which is doubly constrained, being flanked by the slab and the pore on one side and the second $3_1$ knot on the other. The second knotted component, pushed by the first one, eventually reaches the free DNA end and thus becomes untied. From this point, the dynamics proceeds with the remaining $3_1$ knot, which reaches about the same size at the end of the simulated trajectory as the isolated $3_1$ knot, about 200 bp (see also Figure 6b).

We conclude that knots in pinned DNA chains can evolve substantially, expanding and becoming untied over timespans comparable to the entire unzipping process at $f = 100$ pN.

### Knot evolution during unzipping

The above dynamics is qualitatively modified when the pinning constraint is removed, and the DNA is forced to unzip by the driven translocation through the narrow pore.

The middle column of Figure 6a is for $f = 50$ pN. In the $3_1$ case, the $n(t)$, $n_1(t)$, and $n_2(t)$ traces are overall parallel, with $n_1$ staying close to $n$ at all times. These facts indicate that the $3_1$ knot remains close to the pore entrance throughout unzipping and maintains its initial moderately tightened state ($l_k \sim 150$ bp) as (from the relative 'perspective of the *cis* chain') it slides along the dsDNA contour at approximately constant velocity.

For the $4_1$ and $3_1\#3_1$ cases, the knots remain close to the pore entrance, and their lengths slightly increase over time, albeit to a lesser extent than for the pinned case, with the $4_1$ reaching $l_k \sim 200$ bp before escaping and $3_1\#3_1$ reaching $l_k \sim 120$ and $l'_k \sim 150$ bp for its prime components (Figure 6b).

Increasing the force to $f = 100$ pN introduces radical changes to knot evolution and sliding dynamics, as seen in the rightmost plots of Figure 6a. The $3_1$ knot exhibits a substantial tightening at the pore entrance, and so does the first $3_1$ component of the composite knot. Both values reach a stationary value of $l_k \sim 25$ bp (Figure 6b). Instead, the length of the second component of the composite knot appears to be only modestly affected, with $l'_k$ fluctuating over values of $\sim 120$ bp. Interestingly, the length of the $4_1$ knot also decreases with time, going from 200 nucleotides at $t = 0$ to 70 at $t = 3 \cdot 10^6 \tau_{MD}$ (Figure 6b), but never reaching the tightness observed at the late translocation stages of the $3_1$ knot.

Finally, at 150 pN, the lengths of the $3_1$ knot and the first $3_1$ component of the composite knot both reach a similar asymptotic $l_k \sim 25$ bp value as the ones of 100 pN, but at a much faster pace (Figure 6b). At this force, the $4_1$ knot can become tighter than at 100 pN, reaching an asymptotic value of $l_k \sim 30$ bp (Figure 6b).

The results clarify that the two dynamical regimes discussed for Figure 3b are directly connected to the degree of tightness of the knot. In fact, the $n(t)$ traces for $f = 100$ pN of Figure 6a indicate that unzipping of the chains does not proceed at a constant pace but progressively slows down. The latter occurs in correspondence with the knot length reduction, conveyed by the close approach of the $n_1(t)$ and $n_2(t)$ curves.

The slow down, as well as its dependence on the applied force and knot type, is analogous to the topological friction found in translocating knotted chains without unzipping, as observed in general polymer models in Rosa *et al.* (2012) and Suma *et al.* (2015), and here confirmed for dsDNA, see Figure 1. Similarly to these cases, the knot slows down the process but does not necessarily halt it entirely, as the chain can still slide on its knotted contour unless the dynamics is jammed by extreme knot tightening. The degree of tightening and, in turn, the associated hindrance depends on the applied force and the knot characteristics, which can change how the tension force propagates along the chain on the *cis* side.

### Conclusions

We used molecular dynamics simulations to study the nanopore unzipping of knotted DNA. In our study, we considered dsDNA filaments of about 500 bp prepared with different types of prearranged moderately tightened knots, namely the unknot (the trivial knot), $3_1$, $4_1$, and $3_1\#3_1$ knots. The filaments were unzipped by pulling one single-stranded terminus through a narrow pore at three different forces, $f = 50$, 100, and 150 pN. The progress of the unzipping process was characterized by analyzing the temporal traces of the number of translocated (hence unzipped) nucleotides and by tracking the position and length of the knotted region along the DNA contour.

The comparative analysis of the unzipping process across the considered knot types and forces enabled us to establish three main results. First, the DNA unzipping process at sufficiently low forces is virtually unaffected by the presence of knots. In fact, at $f = 50$ pN, the translocation traces of all three knot types were practically superposable to those of unknotted DNAs. Second, increasing the force to $f = 100$ and 150 pN caused knotted DNAs to unzip significantly more slowly and heterogeneously than unknotted ones. The highest hindrance was observed for $4_1$-knotted filaments, whose average unzipping at $f = 150$ pN was four times slower than the unknot. The corresponding dispersion of unzipping times was also substantial, accounting for a three-fold time difference between the slowest and fastest trajectories out of a set of five. Finally, analyzing the knotted DNA structure close to the pore revealed that the observed hindrance to unzipping involves at least two concurrent mechanisms: (i) the topological friction arising from the DNA chain sliding along its tightly knotted contour and (ii) the friction caused by the newly-unzipped *cis* DNA strand wrapping around the double-stranded DNA region between the knot and the pore.

The above results have implications in various physical and biological contexts. Because knots are statistically inevitable in sufficiently long DNA filaments, clarifying the impact of such forms of entanglement on how DNA unzips is relevant for polymer physics, particularly for developing predictive models for the complex force-dependent response of such processes. From the applicative point of view, the system and results discussed here could be used in prospective nanopore-based single-molecule unzipping experiments on long (hence knot-prone) DNAs, from interpreting the ionic current traces to designing such setups. Finally, DNA nanopore unzipping can be regarded as a gateway to elucidating the physical processes occurring *in vivo*, where genomic DNA is unzipped and translocated by various enzymes. It would thus be interesting to extend future considerations to DNA lengths and force regimes that match those relevant for *in vivo* DNA transactions as closely as possible, where molecular crowding may also play a role.

## Model and numerical methods

We used a coarse-grained model of DNA, oxDNA2 (Ouldridge *et al.*, 2010, 2011; Snodin *et al.*, 2015), to simulate double-stranded DNA filaments of about 500 bp. Each nucleotide is treated as a rigid body with three interaction centers. The potential energy describing the interactions between nucleotides accounts for the chain connectivity, stacking effects, excluded volume interactions, twist-bend coupling, base pairing (with sequence-averaged binding interactions), and screened electrostatic interactions. The system was evolved with Langevin dynamics simulations using the LAMMPS simulation package (Henrich *et al.*, 2018; Thompson *et al.*, 2022). The temperature was set to $T = 300$ K, and the monovalent salt concentration defining the Debye–Hueckel potential was set to 1 M NaCl, within the range adopted *in vitro* nanopore experiments. Other model parameters were set to the default values of the LAMMPS oxDNA2 implementation, except for the damp parameter, which was increased to 5 as in Suma *et al.* (2023) to reduce inertial effects at the largest used forces. We used a timestep of 0.01 $\tau_{MD}$, with the longest simulation lasting $3.5 \times 10^6 \tau_{MD}$, where $\tau_{MD}$ is the characteristic simulation time.

The DNA strands have excluded volume interactions with a slab with an embedded cylindrical pore; see SI of Suma *et al.* (2023) for the potential. The pore length (slab thickness) is 8.52 nm. The nominal pore diameter was set to 1.87 nm (narrow pore) and 4.25 nm (wide pore) for translocations with and without unzipping of the double helix. The initial setup used in both situations, described hereafter, is the same. Note that 1.87 nm is a diameter sufficient to allow only a single ssDNA strand to pass at a time inside the pore. Given that the thickness (steric repulsion range) of the nanopore is 0.95 nm, the net diameter of the pore is about 1 nm, which is comparable, for instance, to the width of the lumen of biological nanopores used for unzipping, see for example, the MspA protein with a constriction of the order of $\sim 1$ nm (Bhatti *et al.*, 2021). Instead 4.25 nm is sufficient to allow a dsDNA strand to pass, but not a knot, which is necessarily composed of $\geq 3$ strands and hence bound to remain in the *cis* side of the pore.

To produce the initial conformation, we used an analogous scheme to Suma *et al.* (2015): we first employed a Monte Carlo scheme to sample equilibrated configurations of coarse-grained semi-flexible chains with thickness, contour length, and persistence length corresponding to double-stranded DNA filaments of 500 bp. At the front of the chain, a tightened knot was attached of three different types, $3_1$, $4_1$, $3_1\#3_1$, taken from simulations of Suma *et al.* (2015), and long about 50 bp. For the $0_1$ unknotted case, we did not add anything.

The knotted terminus was then attached to a 40-base lead already threaded through the pore. The configuration was subsequently relaxed using an intermediate fine-grained model (see Suma and Micheletti (2017)) for the specifics, by pinning one nucleotide inside the pore. During this relaxation, the initially tightly knotted components expand to about 150 bp to lower the bending energy. The conformation was then mapped to the oxDNA2 representation of double-helical DNA with the tacoxDNA package (Suma *et al.*, 2019), with the lead inside the pore mapped into a single-stranded DNA. The whole chain was again briefly relaxed by pinning one nucleotide inside the pore and letting the system evolve for a time span of $200\tau_{MD}$. Translocation was driven by a longitudinal force, $f = 50$, 100, 150 pN, acting exclusively on the DNA segment inside the pore and equally distributed among the nucleotides in the pore. This technical expedient is adopted to keep the driving force constant.

The relaxed filaments were translocated and unzipped by pulling the ssDNA stretch inside the pore with a constant total force, $f$.

A resulting initial conformation is shown in Figure 2a. At variance with Suma *et al.* (2023), here we show the translocation process for this configuration instead of unzipping the first 200 bp bases, as our main interest is to study the knot positioning and effects. Five different Monte Carlo-generated configurations were used for each topology, and their sequence composition was also randomly picked at the oxDNA fine-graining step. The resulting conformations for the unknot and the three knot types are displayed in Figure 2b during translocation.

Detection of knots was carried out using the software Kymo-Knot (Tubiana *et al.*, 2018). From a mathematical point of view, knots are rigorously defined only for circular chains. Accordingly, to establish the knotted state of an open chain, it is necessary to close it into a ring by bridging its terminals with a suitable auxiliary arc (Tubiana *et al.*, 2011). This step was carried out with the so-called minimally interfering closing procedure, which selects the auxiliary arc that adds the least possible entanglement to the open chain. After closure, the knotted state of the chain is established using the standard Alexander determinants. This way, we assign a definite topological state to each configuration sampled in the MD trajectory and select the DNA nucleotide indexes that delimit the knotted region (further reducing the polymer region would result in not being able to detect the knot). For prime knots, these correspond to indexes $n_1$ and $n_2$, while for composite knots, they correspond to indices $n_1$ and $n_4$ (Figure 6a). The prime components within a composite knot were identified by using a bottom-up search.

**Open peer review.** To view the open peer review materials for this article, please visit http://doi.org/10.1017/qrd.2024.26.

**Acknowledgements.** This study was funded in part by the European Union – NextGenerationEU, in the framework of the PRIN Project 'The Physics of Chromosome Folding' (code: 2022R8YXMR, CUP: G53D23000820006) and by PNRR Mission 4, Component 2, Investment 1.4_CN_00000013_CN-HPC: National Centre for HPC, Big Data and Quantum Computing – spoke 7 (CUP: G93C22000600001). The views and opinions expressed are solely those of the authors and do not necessarily reflect those of the European Union, or can the European Union be held responsible for them.

**Competing interest.** The authors declare no competing interest. All authors have contributed to this submission.

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
