## [Reviewer Report]

The authors used molecular dynamics simulations and a coarse-grained DNA model to study the effect of unzipping of knotted DNA filaments by a single-stranded DNA nanopore. They observed that knots of different types do not affect unzipping at low forces. However, heterogenous and slow unzipping trajectories are observed at high forces, especially for the complex 41 and 31 # 31 knots. Simulations showed that topological friction and entanglement of the unzipped DNA are the microscopic origin of this hindrance.

Specific comments

1. The reviewer agree that since knots are statistically inevitable in sufficiently long DNA filaments, studying the impact of knots on DNA unzipping are of relevance in biology, polymer physics and nanopore technology.

2. Figure 1 is from which reference?

3. The authors mentioned the relevance of studying unzipping of a dsDNA with a knot. Can the authors provide few examples of a biological process with this specific feature that is being proven experimentally?

4. The reviewer find interesting the fact that mimicking the action of enzymes doing DNA unzipping with a nanopore can provide insight on the effect of entanglement and topological friction during the DNA translocation inducing unzipping. The authors choose a pore size of 1.87nm. Is it comparable to DNA motors performing unzipping? If so, can the author compare this value to the ones determined by structural methods? If not, why the authors used the specific value?

5. The authors did not observe significant eﬀect of knots on DNA unzipping at forces of 50pN. Presumably, this is also be the case for forces bellow 50 pN. However, DNA motors studied so far by various single molecule methods cannot apply such force or higher. Is it possible that in the presence of crowding, the unzipping becomes more force-sensitive? Can the authors test this idea?

6. For Figure 3, the decrease in the velocity due to the tightening of the knot near the pore entrance would become more evident to the reader if the authors can add a Figure showing how exactly at the inflection point, the knot is tight as show in Figure 5.

7. It would be very useful if the authors can put explicitly the dimensions of the pore in Figures, so the reader does not have to wait till the end the article to find out.

8. Why the newly unzipped cis DNA strand wraps around the dsDNA? Can the authors provide an explanation?

9. Why 1M NaCl? This is non-physiological.

10. Could the authors be more explicit about the meaning of 5 as friction coefficient?

---

## [Reviewer Report]

The manuscript explores the nanopore-driven unzipping of knotted DNA using molecular dynamics simulations and coarse-grained modeling, focusing on how DNA knots affect the unzipping process under varying driving forces. It demonstrates that at low forces, knots do not significantly hinder unzipping. In contrast, at high forces, knotted DNA unzips more slowly and heterogeneously compared to unknotted DNA. The hindrance arises from topological friction and entanglement of the unzipped DNA with the knotted region. The study offers valuable insights into the mechanisms governing DNA knot unwinding and translocation. However, several key issues need to be addressed before acceptance:

1) Authors should provide data to substantiate the torque claim: “These wrappings originate from unwinding the unzipped DNA, which imparts a torque to the translocating molecule that accumulates at sufficiently high force when torsional stress is introduced faster than the relaxation dynamics can dissipate it.”

2) Discuss how high-force application might alter DNA structure and how these structural changes affect the unzipping of knots.

3) Provide a clear justification for selecting the oxDNA2 model. Discuss its suitability for studying DNA unzipping under the specified conditions.

4) Ensure that figure legends are more descriptive, explaining what each analysis represents and how it was performed. For instance, clarify how parameters like Ik and Ik’ are defined and calculated.